# Single-cell imaging of α and β cell metabolic response to glucose in living human Langerhans islets

Fabio Azzarello[1,3], Luca Pesce [1,3], Valentina De Lorenzi[1], Gianmarco Ferri[1], Marta Tesi[2], Silvia Del Guerra[2], Piero Marchetti[2] & Francesco Cardarelli [1✉]

Here we use a combination of two-photon Fluorescence Lifetime Imaging Microscopy (FLIM) of NAD(P)H free/bound ratio in living HIs with post-fixation, immunofluorescence-based, cell-type identification. FLIM allowed to measure variations in the NAD(P)H free/bound ratio induced by glucose; immunofluorescence data allowed to identify single α and β cells; finally, matching of the two datasets allowed to assign metabolic shifts to cell identity. 312 α and 654 β cells from a cohort of 4 healthy donors, 15 total islets, were measured. Both α and β cells display a wide spectrum of responses, towards either an increase or a decrease in NAD(P)H free/bound ratio. Yet, if single-cell data are averaged according to the respective donor and correlated to donor insulin secretion power, a non-random distribution of metabolic shifts emerges: robust average responses of both α and β cells towards an increase of enzyme-bound NAD(P)H belong to the donor with the lowest insulin-secretion power; by contrast, discordant responses, with α cells shifting towards an increase of free NAD(P)H and β cells towards an increase of enzyme-bound NAD(P)H, correspond to the donor with the highest insulin-secretion power. Overall, data reveal neat anti-correlation of tissue metabolic responses with respect to tissue insulin secretion power.

[1] NEST Laboratory - Scuola Normale Superiore, Piazza San Silvestro 12, Pisa, Italy. [2] Department of Clinical and Experimental Medicine, Islet Cell Laboratory, University of Pisa, Pisa, Italy. [3] These authors contributed equally: Fabio Azzarello, Luca Pesce. ✉email: francesco.cardarelli@sns.it

Blood glucose levels are finely tuned in a narrow range by the regulated secretion of the pancreatic islet hormones insulin and glucagon in response to fluctuations in glucose and other metabolites concentration[1–4]. A cascade of highly regulated biochemical processes connects glucose influx to hormone secretion in specialized cells within the islets of Langerhans of the mammalian pancreas, i.e., the α and β cells[2–6]. Such regulation is progressively lost in diabetes, leading to onset and worsening of the disease[7,8]. Given the importance of these processes for systemic glucose homeostasis and for our comprehension of their mis-regulation in pathology, non-invasive and fast strategies capable of monitoring quantitatively α and β cells' metabolic responses in living islets are highly desirable. Typically, the efficiency of the overall process can be described by monitoring the amount (and timing) of secreted hormones with respect to the amount of the administered stimulus by quantitative ELISA-based assays. These assays provide a useful experimental platform to monitor the general responsiveness of α and β cells to stimulation and the possible alteration of their behavior in the pathological condition[9], although they average out the biochemical processes that occur in between stimulation and secretion. On the other hand, the possibility to monitor directly molecular processes within living α and β cells by using suitably labeled target molecules (e.g., by introducing genetically encoded fluorescent proteins) and optical microscopy is challenged by the need of not to alter the chemical identity and endogenous stoichiometry of the biochemical components involved in the metabolic response of these cells[10]. In this regard, in the pioneering works by the Piston's Group, two-photon excitation microscopy was used to image and quantify the intrinsic NAD(P)H auto-fluorescence from both dispersed mouse-derived α/β cells and intact pancreatic islets under glucose stimulation[11–13]. The authors were able to measure the change in whole-cell NAD(P)H levels in response to glucose stimulation and spatially resolve NAD(P)H signals from the cytoplasm and mitochondria of cells[3]. Using this intensity-based approach, additional studies were conducted to evaluate islet metabolic response to several stimuli, both in mouse[14] and human[15] systems. However, NAD(P)H intensity-based measurements may contain artifacts due to the heterogeneity of fluorophore concentration and to the differing quantum yields of NAD(P)H species[16]. To bypass this limitation, Fluorescence Lifetime Imaging Microscopy (FLIM) emerged as a potential useful tool as it is minimally affected by cell absorption/scattering and/or fluctuation in excitation intensity and, in addition, it can selectively discriminate the free and protein-bound forms of NAD(P)H molecules[17,18]. So far, FLIM on NAD(P)H, especially in combination with the phasor-based graphical and fit-free data-analysis approach[19–22], found widespread use in the study of cell metabolism and its possible alteration during processes such as disease progression, differentiation, cell fate, cell division[21–24]. For what concerns islet metabolism, phasor-FLIM analysis of NAD(P)H species was used to monitor the metabolic status of both mouse- and human-derived intact islets under different stimuli, including pulsed glucose stimulation. For instance, in the pivotal works by Gregg and collaborators on aging and obesity, phasor-FLIM of NAD(P)H signal was used in both mouse- and human-derived intact islets under different stimuli[25,26]. The authors, however, interpreted the observed metabolic responses as deriving exclusively from β-cells because of their dominance in the islet, effectively forgoing to identify α cells and quantify their fractional contribution to overall islet response. Similarly, Haythorne and collaborators measured islets from both healthy and diabetic mouse models using phasor-FLIM of NAD(P)H signal but attributed the obtained metabolic response again to the (dominant) contribution of β cells[27]. Of note, Wang and collaborators recently set out

to discriminate the specific responses of α and β cells to glucose stimulation[28]. The authors started from dispersed primary mouse α and β cells and used a combination of phasor-FLIM of NAD(P)H signals to study metabolism and immunofluorescence to identify cell types[28]. Interestingly, a neat discordant response to glucose from the two types of cells was observed, with α cells shifting towards a decrease in the bound/free NAD(P)H ratio and β cells towards a concomitant increase of the same ratio[28]. The authors extended similar conclusions to α and β cells measured within the intact mouse islet, where cell-cell interactions and possible paracrine effects persist, but for cell-type identification they relied in this case mainly on the assumption that α cells are located in the periphery and β cells in the core of the mouse islet[28]. Finally, the authors measured human-derived islets from donors with and without T2D but, due to the more dispersed distribution of α and β cells in the human islets compared to mouse islets, they were unable to reliably identify cell types and thus characterize their specific response to glucose[28]. Here, after validation of phasor-FLIM by using an immortalized cellular model of β cells (i.e., Insulinoma 1E cells, hereafter INS-1E cells, Fig. S1), we addressed the specific metabolic responses of α and β cells from living human-derived Langerhans islets by using an experimental protocol consisting of three main phases, namely: (i) NAD(P)H auto-fluorescence lifetime imaging in live islets at both low (2.2 mM) and high glucose (16.7 mM), with subtraction of the lipofuscin intrinsic signal, typical of human pancreatic islets (ii) islet fixation and immunostaining using antibodies against glucagon and insulin to identify single α and β cells, respectively, (iii) matching of immunofluorescence and live-islet acquisitions to extract single-cell information from phasor-FLIM data. By this approach, 312 α cells and 654 β cells were identified from 15 total islets (HI) of 4 non-diabetic human organ donors, and analyzed. While demonstrating the effectiveness of an optical-microscopy-based protocol to measure the specific responses of α and β cells in a living human Langerhans islet, present results suggest an emergent relationship between the optical signature of HIs and their insulin secretion power of potential biomedical/clinical interest.

## Results

**Phasor-FLIM on NAD(P)H in live islets: experimental approach and system calibration.** The experimental approach is schematically represented in Fig. 1. It consists of three main phases, namely: (i) NAD(P)H auto-fluorescence lifetime imaging at a focal plane in live islets at both low (2.2 mM) and high glucose (16.7 mM), with subtraction of the lipofuscin intrinsic signal, to produce whole-islet metabolic data (Fig. 1a–d), (ii) islet fixation and immunostaining using antibodies against glucagon and insulin to identify single α and β cells and achieve single-cell and cell-type resolution (Fig. 1e, f), (iii) spatial matching of immunofluorescence and live-islet acquisitions to extract single-cell information from phasor-FLIM metabolic data (Fig. 1g). To start, we calibrated our system using an immortalized cell line that is a recognized model of β cells: Insulinoma 1E (INS-1E) cells (Fig. S1; experimental details can be found in the "Methods" section). Results on INS-1E cells are reported in Fig. S1: in brief, glucose stimulation induces a neat shift of INS-1E cells towards an increase in bound/total NAD(P)H ratio (hereafter referred to as "shift towards bound NAD(P)H"; conversely, a shift towards a decrease in bound/total NAD(P)H ratio will be referred to as "shift towards free NAD(P)H"), in keeping with previous results by some of us[29] and others[28]. The bound/total NAD(P)H ratio is calculated from the barycenter of the phasor using Eq. 3 in "Methods" and Fig. S2. At this point, we started experiments on HIs. Before imaging, stabilization of HIs was achieved by using

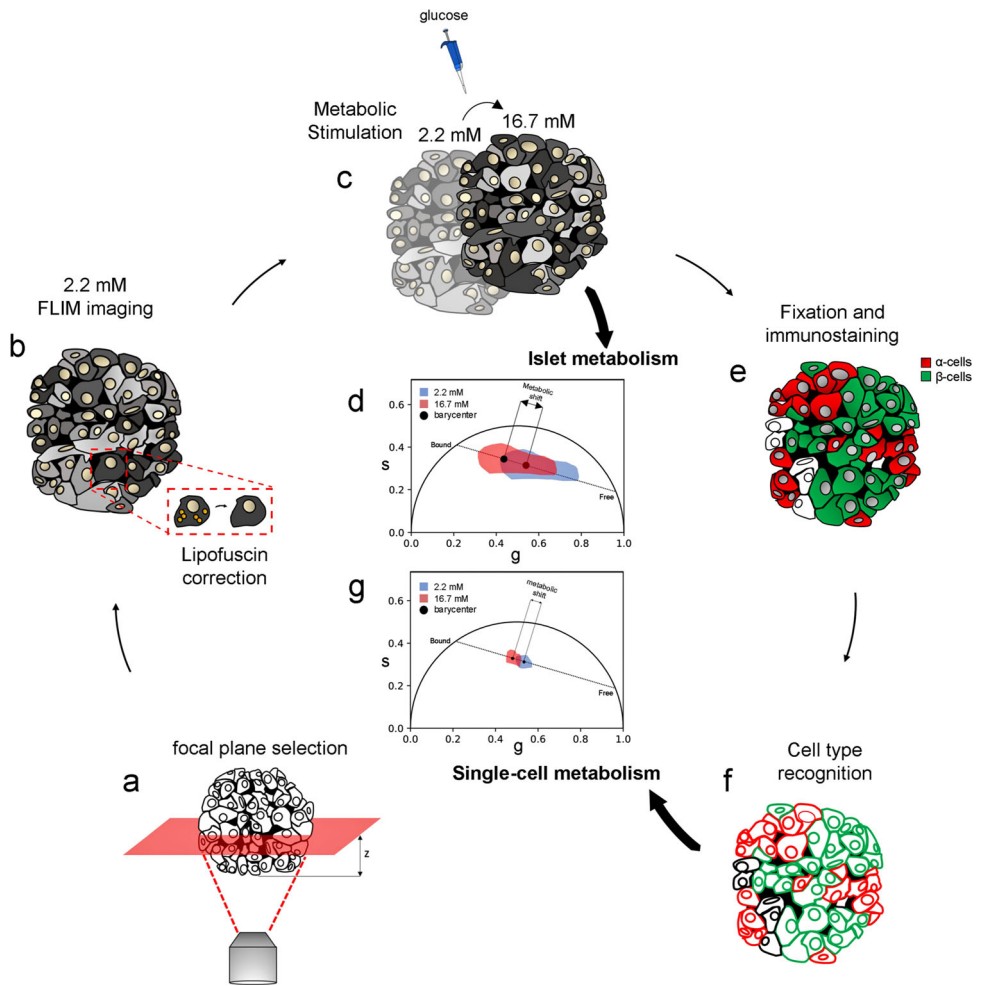

**Fig. 1 Experimental pipeline optimized for investigating human pancreatic islets by FLIM. a** Human pancreatic islet imaging requires immobilization (using agarose gel, see Methods) and correct focal plane positioning, due to the high scattering properties and absorption process of the tissue by chromophores and pigments (i.e., lipofuscin), which prevent deep light penetration. At the selected plane (**b**), tissue auto-fluorescence lifetime was measured at 740 nm and collected in the 420–460-nm range. The first measurement was performed at 2.2 mM glucose concentration and subtraction of lipofuscin auto-fluorescence was performed to univocally assign the remaining signal to NAD(P)H (see methods for a detailed description regarding the correction). **c** Stimulation was performed by increasing glucose concentration from 2.2 to 16.7 mM: 3 minutes after the stimulation a second FLIM image was acquired and corrected for lipofuscin signal as well. At this level, the phasor barycenter from the two FLIM images can be calculated and their coordinates along the metabolic axis define the so-called metabolic shift and its magnitude. **d** To obtain single-cell metabolic data, the islet was fixed and then incubated with anti-glucagon and anti-insulin antibodies to recognize α and β cells, respectively. Within this step, the islet was imaged again at the same focal plane used for metabolic imaging, but in a standard 2-channel immunofluorescence experiment (α-cells in "red" and β-cells in "green"). **e** FLIM images at 2.2 and 16.7 mM are compared with the immunofluorescence one with subsequent cell type recognition and single-cell ROI creation in both FLIM images (**f**). At this point, each recognized cell has an associated phasor plot. **g** Single-cell metabolic shifts are calculated in the same way as whole-islet ones, namely by determining the barycenter coordinate shift along the metabolic axis from 2.2 to 16.7 mM.

agarose at 1% in SAB medium, which allowed reducing islet mobility throughout the imaging/stimulation experiments while preserving islet viability (see "Methods" for further details). Islet auto-fluorescence was elicited by 2-photon excitation at 740 nm and collected in the 420-460-nm emission range. The optical-sectioning depth was chosen in the 10–15 μm range from the glass and kept constant for all the acquisitions: this choice was dictated by the observation that light penetration (and consequently imaging) deteriorates with depth, due to absorption/scattering by the tissue. As control, the efficiency of light penetration in a live islet stained with Hoechst was compared with the same experiment conducted after islet fixation and clearing with the TDE/PBS solution[30,31] (Fig. S3, see also "Methods" for further details). At the selected optimal depth, FLIM measurements were performed and analyzed using the phasor approach, allowing the lifetimes measured at each pixel in the image to appear in the

corresponding region of the phasor plot. (Fig. 2a, b). As can be appreciated from the phasor plot reported in Fig. 2b, a rather broad distribution of lifetimes is measured in the live-islet optical section, positioned along the expected "metabolic segment". Contrary to INS-1E cells, in this case a main correction needs to be made before assigning the measured lifetimes univocally to NAD(P)H. As suggested in previous works, in fact, the auto-fluorescence signal coming from lipofuscin-containing aggregates, which are typical of α and β cells in HIs, can potentially interfere with NAD(P)H imaging and lifetime analysis[29]. Indeed, at the imaging conditions selected here, intense punctuate signal was detected in the majority of cells (Fig. 2c). Interestingly, considering the 4-donors cohort used in this study, the overall amount of lipofuscin detected by imaging was found to correlate with donor age and Body Mass Index (BMI) (Fig. S4a, b; refer also to Tabs. S1 and S2 for donor-related parameters), a result in

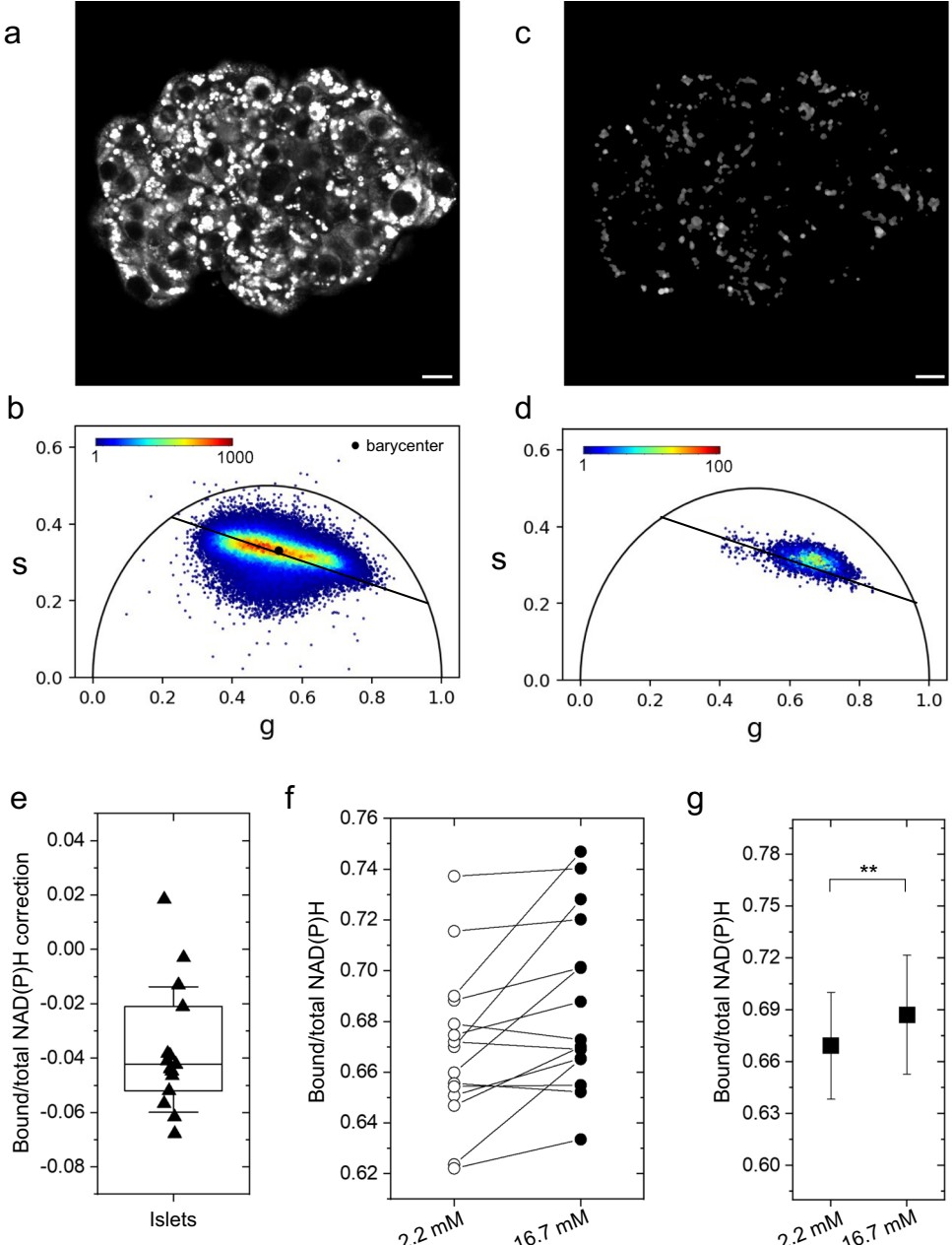

**Fig. 2 FLIM imaging and lipofuscin subtraction in living human islets. a**, **b** Likewise immortalized cells, FLIM images of living human islets depict the distribution of fluorescence intensity with each pixel having a corresponding point in the phasor plot. The islet metabolic status can be modeled using the phasor barycenter and its movement along the metabolic axis upon increasing glucose concentration from 2.2 and 16.7 mM. Before assigning signals to NAD(P)H, the contribution of lipofuscin bodies (**c**) needs to be subtracted. Lipofuscin signal (**d**) potentially interferes with barycenter calculus, causing barycenter displacement towards free NAD(P)H condition ($n = 15$ islets) (**e**), yielding lipofuscin correction a necessary task for the correct identification of islet metabolic state. Considering the islet as a whole, metabolic response for glucose stimulation ($n = 15$ islets) displays mild to moderate bound-shift (**f**), yielding a statistically significant bound-shift on average ($n = 15$ cells) (**g**), presumably due to the higher amount of β cells in the islet which thus have higher contribution in barycenter calculus. At this point, no cell type recognition has been performed yet. Statistical difference obtained using Paired Samples Wilcoxon Test, with significance $\alpha = 0.05$ (**$P < 0.01$). Scale bar: 20 μm.

keeping with previous systematic studies conducted on human islets[30]. Lipofuscin was subtracted by introducing a filter based on the intensity (see "Methods" for further details). This correction eliminates a lifetime component with a peculiar location in the phasor plot, i.e., close to free NAD(P)H on the metabolic segment (Fig. 2d), that would affect the estimate of the phasor barycenter and the calculation of the bound/total NAD(P)H ratio (Fig. 2e). Yet it must be noted that, since the amount of lipofuscin does not change after stimulation, its subtraction has a negligible effect on

the calculation of the metabolic shift (Fig. S4c, d). At this point, the signal left after lipofuscin subtraction was attributed to NAD(P)H auto-fluorescence, and the basic experiment of glucose stimulation was performed by acquiring two consecutive FLIM images for each islet: the first at low glucose concentration (2.2 mM, after incubation in the SAB solution for 45 min), the second after exposure of the islet to high glucose concentration (16.7 mM). All stimulated islets were measured within 15 min from glucose administration. The corrected bound/total NAD(P)

H ratios from all measured islets ($N = 15$), before and after stimulation, are reported in Fig. 2f. Overall, glucose stimulation induced a metabolic shift towards bound NAD(P)H in the majority of the islets. This average result is in keeping with previous observations on human islets[26,27] but, as discussed above, it does not contain any information on the specific contribution from α and β cell populations in the islet.

**Islet fixation and immunostaining to identify α and β cells**. To retrieve information on cell identity, islets were fixed and stained using anti-glucagon and anti-insulin antibodies readily after the FLIM acquisition at 16.7 mM glucose concentration (Fig. 3a). After the fixation/staining step, particular attention was paid to finding the same islet optical section used for metabolic imaging. As an exemplary case, the same islet of Fig. 2 is reported also in Fig. 3a, b to highlight that antibody-based identification of α and β cells was conducted in the same optical section used for NAD(P)H auto-fluorescence imaging. Each FLIM image was compared with the immunostained one in order to identify cell type (i.e., α or β) and cell shape; this latter was exploited to draw single-cell ROIs (highlighted by green/red contour lines in Fig. 3b). With this information gained, single-cell metabolic shifts could be finally extracted from the comparison of FLIM acquisitions at 2.2 and 16.7 mM glucose concentration. In the exemplary case shown in Fig. 3c, single-cell metabolic shifts are reported and classified according to cell type (α cells by red bars, β cells by blue bars) and magnitude (bar height; according to Eq. 4 metabolic shift towards bound NAD(P)H are "positive," those towards free are "negative"). In this particular case, 10 α cells and 49 β cells were identified in the selected optical section of the islet: a markedly heterogeneous spectrum of responses to glucose can be appreciated in both cell types, with metabolic shifts either towards bound or free NAD(P)H. Obviously, information on cell identity and metabolism can be combined together to produce maps which contain the spatial distribution, the magnitude and direction of the metabolic response to glucose of each cell (reported in Fig. S5). The workflow described so far was completed for all the islets included in this study (an additional example is reported in Fig. S6). Worthy of mention, FLIM measurements of metabolic shifts can be used to estimate the number of NAD(P)H molecules involved in the metabolic change, by means of Eq. 5 (see "Methods"). For the islet of Fig. 3, for instance, the highest shifts measured in the two directions are of about 0.06 (cells "36" and "43," Fig. 3c). Assuming an average concentration of NAD(P)H in the cell of approximately 100 μM[31], and an average volume of the cell of 1280 μm$^3$[32], Eq. 5 (see "Methods") leads to an estimate of approximately $4.6 \cdot 10^6$ NAD(P)H molecules which are shifting towards their bound form in cell 36 or towards their free form in cell 43. Interestingly, these FLIM-based estimates are in keeping with previous ones obtained by others using NAD(P)H as test molecule[11]. In addition, it should be kept in mind that what is measured by FLIM are the so-called "fractional intensities" of the two NAD(P)H forms, which are related to their actual molar fractions by means of the quantum yields (QYs) of the pure bound and free species[33]. Taking into account that the average QY for the bound NAD(P)H form is ~8.5 times larger than that for the free NAD(P)H form (estimated by the difference in lifetimes[20,34]), the measured fractional intensities can be corrected and used to extract the molar fractions of free and bound NAD(P)H from FLIM data (Eq. 6). From these calculations, the stationary-state metabolic status appears with a predominance of NAD(P)H in the free form (80% for both α cells and β bells) as compared with NAD(P)H in the bound form (20% for both α cells and β bells). Overall, from the $N = 15$ islets analyzed a total of 312 α cells and 654 β cells

were identified, which yielded a relative abundance of the two types of cells of 32% and 68% for α and β cells, respectively (Fig. 3d) in keeping with expectations for human islets[3,35] (α/β cell proportions for each measured islet are reported in Tab. S3). Interestingly, having at our disposal the identity (i.e., α or β) of each single cell measured allowed us to correlate the amount of lipofuscin detected per single cell to the cell type. As reported in Fig. S4c, it was found that β cells contain substantially more lipofuscin (approximately twofold) as compared to α cells, again in keeping with previous estimates[36]. Most importantly, cell identity was used to extract the average metabolic response of all the α and β cells collected in our experiments: results are reported in Fig. 3e and show an average response of both cell populations (α cells in red, β cells in green) towards bound NAD(P)H. This result apparently contradicts the markedly discordant metabolic trajectories of α cells and β cells observed by Wang and co-workers in disaggregated and intact mouse islets[28], as discussed extensively above. Yet, the population-averaged data reported in Fig. 3e do not exclude the possibility of single-cell metabolic shifts towards free NAD(P)H. Of note, in fact, if metabolic shifts are not averaged together but just classified according to their magnitude and sign (i.e., positive if towards bound NAD(P)H, negative if towards the free form), both α and β cells display metabolic responses in both directions, with a prevalence of cells engaged in a shift towards bound NAD(P)H (i.e., approximately 60% in both cell types) (Fig. S7).

**From single-cell to single-donor analysis**. The similarity in α- and β-cell responses described so far at the population level does not necessarily mean that the two cell types cannot show discordant metabolic trajectories if grouped and analyzed at the islet- or even at the donor-level. To check whether this is the case, α and β cells were first grouped according to the islet from which they originated (Fig. 3f). Each point in the plot now represents the average response of either α (red) or β (green) cells which belong to the same islet. Interestingly, if grouped at this level, the majority of islets (approximately 80% over a total of $N = 15$ islets from $N = 4$ donors, compared to 60% from single-cell data) show an average β-cell-specific metabolic response towards bound NAD(P)H (Fig. 3f, green points). By contrast, only 40% of the islets show α-cell-specific metabolic response in the same direction (Fig. 4a, red points), compared to the 60% calculated from single-cell data. Thus, single-cell data grouped at the islet level suggest a non-random distribution of α- and β-cell metabolic responses among different islets. To further investigate this point, single-islet data were further traced back to the respective donor and correlated to its insulin secretion power, for both α and β cells (red and green data points and linear interpolations in Fig. 4): noteworthy, robust and concordant average responses of both α and β cells towards bound NAD(P)H correspond to the donor with the lowest insulin secretion power (i.e., Donor-1 data points); then, such responses differentially decrease in magnitude (i.e., α-cell responses decrease faster than β-cell ones) in donors with intermediate insulin-secretion powers (i.e., Donor-2 and Donor-3 data points); finally α- and β-cell responses end up to be discordant in the donor with the highest insulin-secretion power (i.e., Donor-4 data points), with α cells shifting towards free NAD(P)H and β cells in the opposite direction, a result in line with what previously observed by Wang and collaborators[28]. Further investigations will be needed to clarify whether these results reflect a regulatory action imparted on α cells by paracrine factors associated to glucose-induced secretion by β cells (e.g., insulin, zinc, GABA, etc.), in addition to glucose itself. Worthy of mention, if α- and β-cell responses from each donor are summed together (i.e., reconstituting the whole-islet metabolic response), marked linear anti-

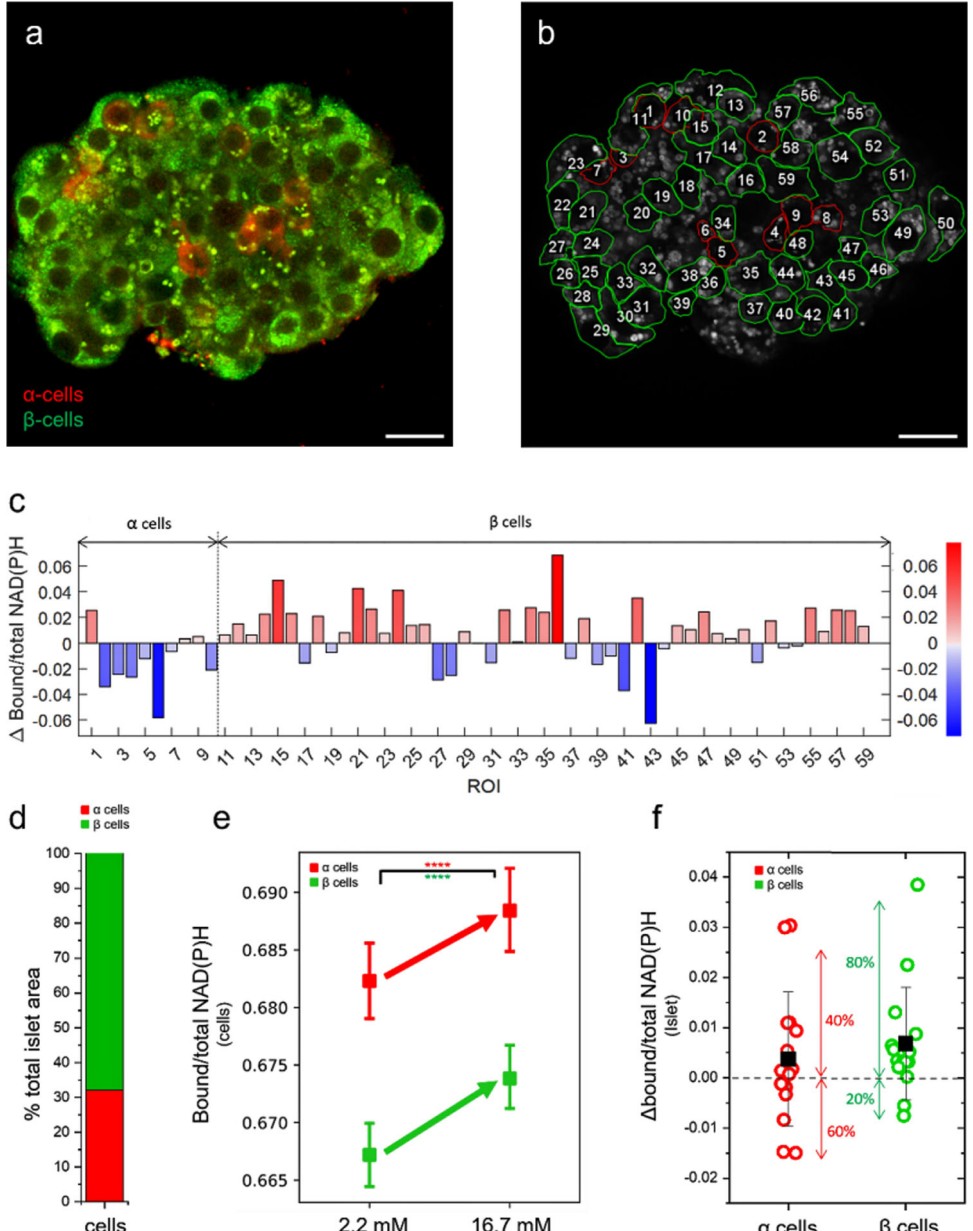

**Fig. 3 From FLIM imaging to single-cell metabolism. a** Immediately after 16.7 mM glucose FLIM imaging, cells are fixed to preserve their shape and structure and then incubated with anti-glucagon and anti-insulin antibodies which in turn recognize α (red) and β -cells (green). **b** Comparison of 2.2 and 16.7 mM FLIM images with immunofluorescence image allow us to recognize cell borders and identity. At this point, single-cell segmentation occurs by tracing ROIs on the two FLIM images (here reported only the 2.2 mM as representative image). Each segmented cell has a unique and equal ROI number in both 2.2 and 16.7 mM FLIM images. Then, single-cell barycenter are calculated, converted in Bound/total NAD(P)H ratio and their difference between 2.2 and 16.7 mM state determines the single-cell metabolic shifts (**c**). Each cell ROI represents a single cell with red bars indicating bound-shift and blue ones free-shift. **d** Considering all the segmented cells, the relative abundance of α (~30%) and β cells (~70%) is in keeping with literature[3] and their average response is bound-shifted for both cell types (**e**), in apparent contrast with literature. **f** If single-cell metabolic shifts are averaged at islet level, α cells ($n = 312$ cells from 15 islets obtained by 4 different donors) show shift towards free NAD(P)H in 60% of islets with the remaining 40% shifting towards bound NAD(P)H; by contrast, for what concerns β cells ($n = 654$ cells from 15 islets obtained by 4 different donors), approximately 80% of the islets display a metabolic shift towards bound NAD(P)H and only the remaining 20% towards free NAD(P)H. Statistical difference obtained using Paired Samples Wilcoxon Test, with significance $\alpha = 0.05$ (****$P < 0.0001$). Scale bar: 20 μm.

correlation with respect to the donor secretion power emerges (Fig. 4, black data points; Pearson's correlation coefficient: −0.997). Interestingly, the 4 donors included in this study are representative of the overall distribution of insulin-secretion indexes typically measured by the ELISA assay[37,38]: the reference gray curve in Fig. 4 is calculated on a total of $N = 17$ donors

(insulin secretion indexes reported in Tab. S1). Again, further studies will be needed to clarify whether this result, even if devoid of information on the two cell types, is potentially revelatory of a marked general relationship between the optical measurement of islet auto-fluorescence lifetime (in the form of a metabolic shift) and the secretion power of the respective donor (and vice versa).

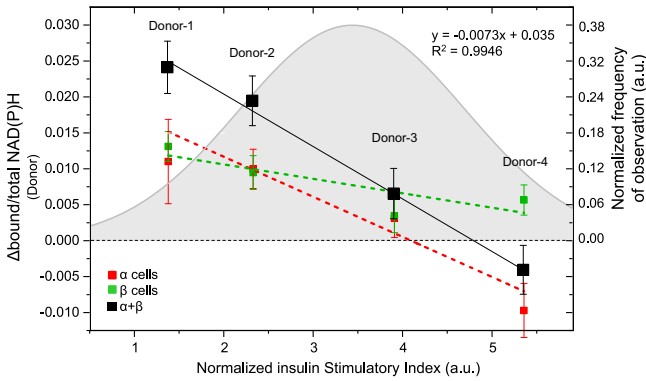

**Fig. 4 coupling of FLIM metabolic data with ELISA-based insulin secretion assays.** Donor-averaged metabolic shift of β cells (green points, $n = 654$ cells from 15 islets obtained by 4 different donors) always display a metabolic shift towards bound NAD(P)H, in accordance with present literature; by contrast, α cells averaged at donor level (red points, $n = 312$ cells from 15 islets obtained by 4 different donors) show a decreasing trend, progressively shifting from "positive" Δbound/total NAD(P)H values to "negative" ones. In addition, the sum of α- and β-cell responses per each donor (black points) shows a markedly linear trend (fitting equation and $R^2$ coefficient reported in the graph legend). The gray curve represents the Gaussian fitting of independently measured (thorough ELISA assays) insulin Stimulatory Index (normalized for insulin content) of $n = 17$ donors, whose detailed description in reported in Tab. S1. Data are presented as mean ± SE.

## Discussion

Here we measured the metabolic response to glucose in 312 α and 654 β cells in 15 human islets from a cohort of 4 healthy donors exploiting fit-free phasor-FLIM analysis of NAD(P)H signals. The metabolic status of a cell was defined by the average position of FLIM data along the so-called "metabolic segment" in the phasor plot, a segment connecting the characteristic lifetimes of the two main forms of NAD(P)H: free and enzyme-bound. Thus, technically, collected data afford a quantitative estimate of the balance of free and bound NAD(P)H in the cell (in this report, analogously to previous ones[28], such balance is expressed by the bound/total NAD(P)H ratio). In our experimental protocol, two consecutive measurements of bound/total ratio are performed, the first at 2.2 mM glucose, the second upon shifting cells to 16.7 mM glucose.

A thorough understanding of cellular responses must inevitably take into account the high complexity of cell biochemistry, in terms of enzyme identity and enzyme expression level, and the differences among cell types, in this case those regarding α and β cells. In fact, variations in the identity and/or expression of enzymes involved in NAD(P)H processing may reflect into variations of the free/bound NAD(P)H balance (and of its response to stimuli). For instance, some of us recently demonstrated that non-secretory cells show indeed a different response to glucose stimulation as compared to INS-1E cells[39]. By the same reasoning, it should be kept in mind that a similar variation in the balance of free and bound NAD(P)H may have different biochemical explanations in different cell types, depending on the peculiarities of cell biochemistry. Worthy of mention, cell biochemistry is not supposed to change during the whole FLIM assay (which lasts few minutes in total); we are prompted to speculate, instead, that the increase in glucose concentration makes cell biochemistry settle down to a different value of the balance of free and bound NAD(P)H. In other words: upon shifting the culturing conditions from 2.2 to 16.7 mM glucose, the same cell

biochemistry rearranges NAD(P)H bound/free ratio to a new value, towards an increase in either bound or free NAD(P)H.

Such reasoning naturally applies to α and β cells, which show specific differences, among others, in glucose handling[40], expression of key metabolic enzymes[41] and enzymatic activity[42]. For instance, the high levels of glycerol 3-phosphate dehydrogenase expression[42] as well as the tight coupling between glycolysis and Krebs cycle[40] in β cells are in line with the well-known commitment of these cells to rapidly increase ATP levels in response to glucose and supports the idea that the measured increase in bound NAD(P)H may reflect activation of the oxidative phosphorylation pathway, as already proposed in the literature[25,28,39]. By contrast, for instance, α cells are expected to use much more the lactate dehydrogenase enzyme (which uses NAD(P)H as cofactor) as compared to β cells[40], and this may reflect into a substantial contribution of this pathway in generating bound NAD(P)H upon glucose stimulation.

In conclusion, it is worth mentioning that reported results have potential implications from both the methodological and biomedical/clinical point of view. Concerning the methodological approach, we demonstrated the effectiveness of an optical-microscopy-based protocol to measure the specific responses of α and β cells in a living human-derived Langerhans islet, thus opening to similar investigations under different conditions of biomedical/clinical interest. Yet, a few lines of future development on this side can be envisioned. In any biophysical measurement, in fact, a compromise has to be found for the resolution achievable in the two main dimensions of interest, the spatial and the temporal one. Concerning the spatial dimension, for instance, it was shown here that light penetration in the tissue is quite inefficient due to the presence of hundreds of cells tightly packed in 3D in the islet. Although using multiphoton infrared microscopy the effective penetration-depth did not allow imaging of the entire islet and, as consequence, a complete single-cell identification and analysis. Preliminary results obtained using tissue-clearing procedures, as reported here in Fig. S3, are encouraging: along with an increase in imaging performances, these procedures are also compatible with metabolic imaging. In fact, Chacko and co-workers recently demonstrated that the lifetime-based NAD(P)H metabolic contrast in a sample is preserved even after chemical fixation, thus opening to the opportunity to generate useful auto-fluorescence signatures from unstained tissues exposed to different stimuli[43]. Of course, an approach based on fixed samples would lose the privileged access to temporal information preventing, for instance, to measure the metabolic response of the same islet in time. On the other hand, concerning the temporal dimension, the present approach exploits live samples to monitor the metabolic shift but then requires fixation for cell identification. To avoid the fixation step, lentivirus-based transduction systems may represent a valuable option. In fact, by allowing the delivery of genetically-encoded fluorescent proteins (FPs)[44], these systems could be used to identify α and β cells by means of specific FP-tagged protein markers. From a biomedical/clinical perspective, we uncovered the optical signature of the metabolic response to glucose of α and β cells in the human islet. These two populations of cells show a markedly heterogeneous spectrum of responses, encompassing metabolic shifts towards either an increase or a decrease of the bound/free NAD(P)H ratio, in apparent contradiction with results obtained in mouse models[28]. However, it was interesting to note that, if single-cell responses are traced back to the respective donor and correlated to its insulin secretion power, emergent collective properties can be highlighted: in detail, robust and concordant average responses of both α and β cells towards bound NAD(P)H correspond to low insulin-secretion power while robust but discordant responses (i.e., α cells shifting towards free and β cells towards bound

NAD(P)H) correspond to high insulin-secretion power. This in turn speaks in favor of a coordinated response of α and β cells in the islet and of a possible role of paracrine factors associated to glucose-induced insulin secretion in modulating the response of the two populations of cells. Then, if α- and β-cell responses are summed together, marked linear anti-correlation with respect to the amount of insulin secreted by each donor emerges highlighting a possible relationship between the FLIM signature of the islet and its insulin-secretion power. Additional investigations, however, will be needed to increase the statistics at the donor level and thus understand how general such relationship is. If confirmed it would have the merit of being independent from cell-type recognition (and thus from tissue fixation) and thus potentially useful to estimate the donor insulin-secretion power from a rather simple optical measurement on a subset of tissue-derived islets.

## Methods

**Human islet isolation and culture.** The pancreases of 4 non-diabetic donors were used for the isolation and study of islets. The characteristics of islet donors are listed in Tabs. S1 and S2. The procedures were approved by the Ethics Committee of the University of Pisa (21 November, 2013, #2615). Islets were isolated by collagenase digestion followed by density gradient purification, as reported in other works[39], before 1 November, 2022. Maintenance culture was at 37 °C and 5% $CO_2$ atmosphere, in M199 culture medium complemented with 10% bovine serum, 100 U/mL penicillin, 100 μg/mL streptomycin, 750 ng/mL amphotericin B, and 50 μg/mL gentamicin, at 5.5 glucose concentration. A complete list of reagents is reported in Tab. S4.

**Cell culture.** INS-1E cells were kindly provided by Prof. C. Wollheim, University of Geneva, Medical Center and cultured in RPMI 1640 medium with 11 mM glucose supplemented with 10% heat-inactivated fetal bovine serum, 100 Units/mL penicillin, 100 μg/mL streptomycin, 2 mM glutamine, 10 mM HEPES, 1 mM sodium pyruvate, and 50 μM β-mercaptoethanol at 37 °C in a humidified 5% $CO_2$ atmosphere.

**Insulin secretion assay.** Insulin secretion was measured as described in other works[39]. Briefly, batches of 15 handpicked human islets were pre-incubated for 45 min with a Krebs' solution containing 3.3 mM glucose, then challenged acutely (45 min) with 3.3 and 16.7 mM glucose. The supernatant was collected and stored at −20 °C. Total insulin content was extracted with an acid ethanol solution and insulin levels were quantified by the Insulin ELISA kit following the manufacturer's protocol (Mercodia AB, Uppsala, Sweden). Insulin stimulation index was calculated as ratio of insulin release at 16.7 mM glucose over release at 3.3 mM glucose, each expressed as percentage of the total insulin content.

**Glucose exposure in imaging experiments.** Whole islets (250-500 μM diameter) were plated on an 8-well coverglass (Sarstedt; 94.6170.802, see Fig. S8). Next, islets were embedded in a solution consisting of 1% agar (%wt/vol) dissolved in SAB (114 mM NaCl, 4.7 mM KCl, 1.2 mM $KH_2PO_4$, 2.5 mM $CaCl_2$, 1.16 mM $MgSO_4$, 20 mM HEPES at pH = 7.4, and 2.2 mM glucose). After agar gelation, islets were equilibrated in a low-glucose buffered solution (SAB) at 37 °C for 45–90 min. After imaging at the low glucose concentration (2.2 mM), human islets were treated with glucose (stock solution 0.5 M) to reach a final concentration of 16.7 mM and imaged after 3–5 min.

**Clearing process.** For the clearing process, live human islets were soaked in 1% agarose and then stained with Hoechst 33342 (dilution 1:1000 in M199 culture medium complemented with 10% bovine serum, 100 U/mL penicillin, 100 μg/mL streptomycin, 750 ng/mL amphotericin B, and 50 μg/mL gentamicin.) for 1 h at 37 °C and 5% $CO_2$ atmosphere. After imaging, the samples were fixed in 4% PFA following three washes in PBS, 5 min each. For the clearing procedure, the fixed specimens were equilibrated with 30% TDE/PBS (v/v) for 30 min and 68% TDE/PBS (v/v) for 1 h and imaged by two-photon microscopy. For the penetration depth analysis, cellular nuclei of two independent islet samples labeled with Hoechst were detected by Fiji using Region Of Interests (ROIs) of 4 × 4 μm (depth 0 μm: 12 nuclei; depth 10 μm: 30 nuclei; depth 20 μm: 36 nuclei; depth 30 μm: 40 nuclei; depth 40 μm: 40 nuclei; depth 50 μm: 40 nuclei). Mean values with associated standard deviation were calculated by Fiji and plotted using Origin (OriginLab Corporation, Origin 2019b).

**Auto-fluorescence imaging.** Live islet imaging by an Olympus FVMPE-RS microscope coupled with a two-photon Ti:sapphire laser with 80-MHz repetition rate (MaiTai HP, SpectraPhysics) and FLIMbox system for lifetime acquisition

(ISS, Urbana Champaign). NADH has been excited at 740 nm and the emission collected by using a ×30 planApo silicon immersion objective (NA = 1.0) in the 420–460 nm range. Calibration of the ISS Flimbox system has been performed by measuring the known mono-exponential lifetime decay of Fluorescein at pH = 11.0 (i.e., 4.0 ns upon excitation at 740 nm, collection range: 400–570 nm). To prepare the calibration sample, a stock of 100 mmol/L Fluorescein solution in EtOH has been prepared and diluted in NaOH at pH 11.0 for each calibration measurement. For each measurement, a 512 × 512 pixels image of FLIM data was collected until 30 frames were acquired. The acquisition time was typically in the order of 1-2 minutes.

**Lipofuscin signal subtraction.** Human islets fluorescent maps present bright spots due to lipofuscin, which is commonly removed by means of an arbitrary intensity threshold because more brilliant than other fluorescent components. In order to obtain a faster and less user-dependent analysis, an automatic signal subtraction algorithm has been developed. First, it calculates the intensity distribution in the FLIM image and, since the lipofuscin bodies pixels can be seen as outliers, it is possible to discriminate lipofuscin by applying the formula:

$$\text{Threshold} = Q_3 + 1.5 \cdot \text{IQR} \qquad (1)$$

where $Q_3$ is the third quartile and $\text{IQR} = Q_3 - Q_1$ is the interquartile range. The already obtained threshold has been used to create a binary mask comprising all pixels below the already calculated threshold and then, by applying the mask on FLIM images, the lipofuscin-containing pixels are excluded from subsequent analyses. The effect of correction is shown in Fig. 2. Though, the algorithm also presents limitations. For example, it is sensitive to illumination gradients caused by tissue scattering, but this is generally prevented by properly choosing the measurement focal plane, otherwise a pre-processing stage can be performed using Fiji or writing customized code using image processing libraries (e.g., scikit-image or OpenCV). Another limitation could be represented by abnormal cells which are almost entirely constituted by lipofuscin, but this was not the case in our experiments.

**Phasor-FLIM analysis.** The metabolic state measured by FLIM is presented as Bound/total NAD(P)H between 0 and 1. This Bound/total NAD(P)H is calculated from the barycenter $(\underline{g}, \underline{s})$ of the phasor:

$$\bar{g} = \frac{\sum_{i=1}^{N} g_i}{N} \quad \bar{s} = \frac{\sum_{i=1}^{N} s_i}{N} \qquad (2)$$

where $g_i$ and $s_i$ are the coordinates of $i$th point and $N$ the number of pixels (e.g., in a 512 × 512 image, $N = 262144$). Conversion of barycenter coordinates $(\bar{g}, \bar{s})$ into Bound/total NAD(P)H occurs by means of the following equation:

$$\text{Bound/total NAD(P)H} = \frac{f_2}{f_1 + f_2} \qquad (3)$$

where $f_1$ and $f_2$ are the fractional intensities of the free and bound NAD(P)H or, namely, the barycenter distances from the two extremes of the metabolic-axis on the universal circle ($\tau_1 = 0.4$ ns and $\tau_2 = 3.4$, see Fig. S2), which in turn represent the characteristic lifetimes of the free and bound forms of NAD(P)H according to present literature[20]. The FLIM-based measurement of metabolic shifts can be defined as the difference between the two relevant conditions measured, i.e., 16.7 and 2.2 mM:

$$\frac{\Delta \text{Bound NAD(P)H}}{\text{total}} = \frac{\text{Bound NAD(P)H}}{\text{total}}\Big|_{16.7\,\text{mM}} - \frac{\text{Bound NAD(P)H}}{\text{total}}\Big|_{2.2\,\text{mM}} \qquad (4)$$

and allows to estimate the number of NAD(P)H molecules involved in the metabolic change $\Delta N_{\text{NAD(P)H}}$ by the equation:

$$\Delta N_{\text{NAD(P)H}} = \frac{\Delta \text{Bound NAD(P)H}}{\text{total}} \cdot [\text{NAD(P)H}] \cdot V_{\text{cell}} \cdot N_A \qquad (5)$$

where [NAD(P)H] is the average NAD(P)H concentration (mol/m³) in the cell, $V_{\text{cell}}$ (μm³) the average cellular volume, and $N_A$ is the Avogadro constant (mol⁻¹). To obtain biologically relevant data, however, correction of the measured fractional intensities needs to be performed taking into account the difference in brightness between the two forms, i.e., bound NAD(P)H is approximately 8.5 times brighter as compared to free NAD(P)H[20,36]. Based on this, Eq. 1 can be rearranged into:

$$\frac{\text{Bound}}{\text{total}} \text{NAD(P)H} = \frac{\bar{f}_2}{\bar{f}_1 + \bar{f}_2} \qquad (6)$$

where $\bar{f}_i$ are the brilliance-corrected values present in Eq. 2, namely:

$$\bar{f}_i = \frac{f_i}{\epsilon_i \cdot \phi_i} \qquad (7)$$

where $\epsilon_i$ is the molar extinction coefficient and $\phi_i$ the quantum yield of the fluorescent specie.

**Immunofluorescence.** After metabolic imaging, human islets were fixed in 4% paraformaldehyde (PFA; stock solution 32% paraformaldehyde, 10× PBS) at room

temperature (RT) for 1 h in gentle shaking, followed by 3 washes with PBS, 5 min each at RT. Then, the fixed samples were permeabilized in 0.1% TritonX-100 in PBS (PBST) for 30–45 min at 37 °C in gentle shaking. After permeabilization, the primary antibodies (mouse anti-Insulin, rabbit anti-Glucagon both diluted 1:500; see Tab. S3) were incubated for 3 days at 37 °C in gentle shaking. After 3 washes for 10 min each at 37 °C with PBST, the human islets were incubated with the secondary antibodies donkey anti-mouse IgG AF 488 and donkey anti-rabbit IgG AF 594 for 24 h at 37 °C in gentle shaking and then, washed 3 times for 10 h each at 37 °C. The stained human islets were then acquired using a two-photon Ti:sapphire laser (Olympus FVMPE-RS) at the excitation light of 800 nm. The insulin-secondary antibody fluorescence signal was collected at 380–570 nm, while the glucagon-secondary antibody signal was collected at 570–680 nm. Microscope parameters (zoom and focus) were maintained equal to the previous acquisitions (auto-fluorescence images), to perform an efficient islet matching. For single-cell identification, auto-fluorescence and immunostained images were paired and α and β cells were manually segmented using Fiji (Tools; ROI Manager). A custom script finally merges ROIs with raw FLIM.R64 files and automatically performing all the already described tasks, i.e., lipofuscin subtraction and metabolic shift calculation.

**Statistics and reproducibility**. Data are presented as mean ± SE. Statistical difference has been assessed using Paired Samples Wilcoxon Test, with significance $\alpha = 0.05$ on Origin software (OriginLab Corporation, Origin 2019b). In total, 321 α cells and 654 β cells have been analyzed which have been obtained from 15 human Langerhans Islets from 4 different donors (additional information are reported on Tabs. S1–S3).

**Reporting summary**. Further information on research design is available in the Nature Portfolio Reporting Summary linked to this article.

## Data availability
The data underlying the findings of this study are available upon requests to the authors.

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

## Acknowledgements

This work has received funding from the European Research Council (ERC) under the European Union's Horizon 2020 research and innovation program (grant agreement No 866127, project CAPTUR3D). P.M. and M.T. receive support from the Innovative Medicines Initiative 2 Joint Undertaking under grant agreements No 115,797 (INNODIA) and 945,268 (INNODIA HARVEST). These Joint Undertakings receive support from the Union's Horizon 2020 research and innovation program and "EFPIA," "JDRF," and "The Leona M. and Harry B. Helmsley Charitable Trust." The authors want to acknowledge Professor Antonio Lucacchini for contributing to the interpretation of the biochemical data.

## Author contributions

F.A. and L.P. performed experiments, analyzed data, and wrote a manuscript draft; V.D.L. performed immunofluorescence experiments and analyzed data; G.F. analyzed data; M.T. and S.D.G. isolated human islets and performed insulin secretion assays; P.M. designed research and wrote the manuscript; F.C. designed and supervised research, analyzed data, and wrote the manuscript.

## Competing interests

The authors declare no competing interests.
