## [Peer Review File · Communications Biology]

Reviewers' comments:

Reviewer #1 (Remarks to the Author):

The paper by Cardarelli et al titled "Single-cell imaging of α and β cell metabolic response to glucose in living human Langerhans islets" is a very well composed paper. It uses metabolic imaging of NADH using phasor based FLIM to look at the system of both types of cells. To me the best part is the difference between the behavior of the populations and how that changes when they are divided by the donor. This is something that is often ignored and this paper shows why that is important. I do have some queries. I may have missed some of that information but I think they are needed.

1. What is the fraction of α and β cells? The authors mentioned that in the previous studies the presence of both types of cells have been ignored. It will be good to get an idea of the proportions.
2. One of my major comments is related to the fraction of NADH. To me it is the fractional intensity of free NADH and not fraction. They are of course related by the actual fractions and their relative quantum yields – but they should not be the same. This basically comes from Figure S1. The linear additivity of phasor is related to fractional intensity and not fractions. The authors should correct this.
3. Again, I absolutely loved the part of separating the different responses of the overall system and how they separated differential responses coming from individual donors.
4. I think I missed it but how do the authors determine lipofuscin? Does the analysis and identification of lipofuscin based on intensity and then creating a mask that removes those points?
5. Minor point – the plots (e.g. Figure 2F) can be presented as box plots with individual cells as scatter – basically a box+scatter. That gives the readers a better idea of the heterogeneity. I think once these questions are answered, the paper can be accepted for publication.

Reviewer #2 (Remarks to the Author):

Azzarello et al

The authors have sought to extend previous studies from other laboratories (refs 27-29) using FLIM imaging of NAD(P)H pools in pancreatic endocrine cells to assess the changes in glycolytic fluxes in alpha and beta cells, including those from humans, in response to glucose challenge. A strength of the report is the use of human islets, and of post hoc identification of alpha and beta cells, allowing these processes to be understood in the most physiologically relevant setting, and as a function of insulin secretory index. However, and unlike earlier studies in mouse islets, the current studies are complicated by the accumulation in human islet cells of lipofuscin which complicates fluorescence signals. Although the findings are potentially interesting I do have some rather fundamental concerns with the approach used.

Major

1. The Phasor approach pioneered by Piston et al is predicated on the assumption that bound/free NAD(P)H is linearly related to the ratio of oxidative phosphorylation (actually the balance of mitochondrial citrate cycle flux and NADH oxidation) to glycolysis in a given cell or subcellular region – as has been assumed in earlier studies. This reviewer would challenge this assumption, at least in beta cells: if the NADH generated at the GP3DH step of glycolysis is efficiently transferred into mitochondria via the glycerol phosphate shunt in these cells, as is likely to be the case given high levels of glycerol 3-phosphate dehydrogenase expression (refs below), then a perfectly parallel increase in both glycolysis and citrate cycle flux would not necessarily generate a parallel increase in free and bound NAD(P)H. Note that glycolysis and oxidative phosphorylation of glucose are tightly coupled in healthy beta cells due to the weak expression of MCT-1 and LDH; lactate production is consequently close to zero (Sekine et al PMID: 8106462; PMID: 9228023). As a result the increased bound : free NAD(P)H measurements might be incorrectly interpreted as a change in glycolysis:ox phos. The situation is somewhat different in alpha cells (references above), and the situation in beta cells is likely affected in type 2 diabetes (eg by upregulation of MCT1, LDHA). This question requires a more careful and critical discussion and interpretation of the current data.
2. Fig. 4. Related to point 1: isn't another interpretation of these data that the glucose-induced

increase in overall metabolic flux (through glycolysis and the TCA cycle) higher in beta than alpha cells? This wouldn't be unexpected given, for example, differences in the glucose-induced increments in ATP between the two cell types (e.g. see PMID: 19008345) and differences in the expression of key metabolic enzymes (PMID: 28443133)?

3. Fig. 5. I do wonder whether the number of subjects here (4) is really enough to draw a robust conclusion about the relationship between glycolysis/ox phos and insulin secretion? Confounding factors (age, sex, BMI, ethnicity etc) of the donors may well play a role? A much larger data set would really be needed for one to have confidence in the conclusion drawn.

Minor

1. Figure 1 and text at the end of the introduction. This would be more logically located in the Results section.

2. Fig 2. This is rather similar to Fig 1 from Ref 28. Perhaps relocate to a supplementary section? Why does the title refer to "alpha/beta" cell lines. INS1E cells are generally considered a beta cell line (vastly more insulin than glucagon expression)

3. Why was 2.2 mM glucose chosen as the basal concentration? Seems low!

4. The Abstract makes a valiant attempt to describe a complex data set but is ultimately confusing. I am not sure that statements like "discordant for high insulin secretion power" will be understood by most readers. Please try to simplify and make more direct.

5. Line 114 – preliminarILY?

6. Decimal points should be "." not "," throughout

Referee expertise:

Referee #1: Multiphoton microscopy, phasor-FLIM

Referee #2: Metabolic imaging, β -Cell Function

Reviewers' comments:

Reviewer #1 (Remarks to the Author):

The paper by Cardarelli et al titled “ Single-cell imaging of α and β cell metabolic response to glucose in living human Langerhans islets” is a very well composed paper. It uses metabolic imaging of NADH using phasor based FLIM to look at the system of both types of cells. To me the best part is the difference between the behavior of the populations and how that changes when they are divided by the donor. This is something that is often ignored and this paper shows why that is important. I do have some queries. I may have missed some of that information but I think they are needed.

First of all, let us thank the Reviewer for his/her positive evaluation of our work and for highlighting the potential impact of having performed data segmentation from the level of single cells to that of donors. We also thank the Reviewer for highlighting a few aspects of the original manuscript that needed further consideration. In the following, for each raised concern, the action taken.

1. What is the fraction of α and β cells? The authors mentioned that in the previous studies the presence of both types of cells have been ignored. It will be good to get an idea of the proportions.

Thanks for highlighting this point. We agree with the Reviewer on the importance of α/β cells proportions. In fact, a relevant side result coming from our cell identification approach is exactly that of having the cell-type proportions under control. To increase the visibility and overall value of these data, we calculated the α/β cells proportions islet by islet and included the results in the revised Fig. 3D

2. One of my major comments is related to the fraction of NADH. To me it is the fractional intensity of free NADH and not fraction. They are of course related by the actual fractions and their relative quantum yields – but they should not be the same. This basically comes from Figure S1. The linear additivity of phasor is related to fractional intensity and not fractions. The authors should correct this.

The reviewer is absolutely right in pointing out that FLIM gives access to the fractional intensity contribution of bound (and free) NAD(P)H, while the actual fractions can be derived only by knowing the respective quantum yields. We understand that confusion on this point could arise from lack of specifications and details in the original manuscript, especially figure S1 (now S2), as pointed out. The figure legend and selected points of the main text were corrected to make this issue clear.

3. Again, I absolutely loved the part of separating the different responses of the overall system and how they separated differential responses coming from individual donors.

Thanks again for highlighting this!

4. I think I missed it but how do the authors determine lipofuscin? Does the analysis and identification of lipofuscin based on intensity and then creating a mask that removes those points?

Yes, lipofuscin has been identified by intensity which threshold has been calculated using Equation 1 (Materials and Methods section), then a mask created and applied to the FLIM map to exclude lipofuscin bodies. To avoid any confusion on this point, more details and clarifications on the procedure were added in the revised text.

5. Minor point – the plots (e.g. Figure 2F) can be presented as box plots with individual cells as scatter – basically a box+scatter. That gives the readers a better idea of the heterogeneity.

Nice suggestion. We implemented it into the revised figures where possible (e.g. fig 2F, now moved to supplementary Fig.1).

I think once these questions are answered, the paper can be accepted for publication.

Thanks for suggesting paper acceptance after the requested revisions. Based on what detailed above, we are confident the revised version is now acceptable for publication

Reviewer #2 (Remarks to the Author):

Azzarello et al

The authors have sought to extend previous studies from other laboratories (refs 27-29) using FLIM imaging of NAD(P)H pools in pancreatic endocrine cells to assess the changes in glycolytic fluxes in alpha and beta cells, including those from humans, in response to glucose challenge. A strength of the report is the use of human islets, and of post hoc identification of alpha and beta cells, allowing these processes to be understood in the most physiologically relevant setting, and as a function of insulin secretory index. However, and unlike earlier studies in mouse islets, the current studies are complicated by the accumulation in human islet cells of lipofuscin which complicates fluorescence signals. Although the findings are potentially interesting I do have some rather fundamental concerns with the approach used.

We would like to thank the Reviewer for his/her critical reading of our work, for stressing the strengths of our report and, finally, for highlighting a few important points that needed further clarification. In the following, for each raised concern, the action taken.

Major

1. The Phasor approach pioneered by Piston et al is predicated on the assumption that bound/free NAD(P)H is linearly related to the ratio of oxidative phosphorylation (actually the balance of mitochondrial citrate cycle flux and NADH oxidation) to glycolysis in a given cell or subcellular region – as has been assumed in earlier studies. This reviewer would challenge this assumption, at least in beta cells: if the NADH generated at the GP3DH step of glycolysis is efficiently transferred into mitochondria via the glycerol phosphate shunt in these cells, as is likely to be the case given high levels of glycerol 3-phosphate dehydrogenase expression (refs below), then a perfectly parallel increase in both glycolysis and citrate cycle flux would not necessarily generate a parallel increase in free and bound NAD(P)H. Note that glycolysis and oxidative phosphorylation of glucose are tightly coupled in healthy beta cells due to the weak expression of MCT-1 and LDH; lactate production is consequently close to zero (Sekine et al PMID: 8106462; PMID: 9228023). As a result the increased bound : free NAD(P)H measurements might be incorrectly interpreted as a change in glycolysis:ox phos. The situation is somewhat different in alpha cells (references above), and the situation in beta cells is likely affected in type 2 diabetes (eg by upregulation of MCT1, LDHA). This question requires a more careful and critical discussion and interpretation of the current data.

2. Fig. 4. Related to point 1: isn't another interpretation of these data that the glucose-induced increase in overall metabolic flux (through glycolysis and the TCA cycle) higher in beta than alpha cells? This wouldn't be unexpected given, for example, differences in the glucose-induced increments in ATP between the two cell types (e.g. see PMID: 19008345) and differences in the expression of key metabolic enzymes (PMID: 28443133)?

As noted by the Reviewer, the two comments are somehow related. We respond to both, as detailed in the following.

First of all, we very much appreciated this comment by the Reviewer as it gave us the opportunity to address a critical point and increase manuscript intelligibility, clearness, and overall impact. First, we do agree with the Reviewer that a weakness of the original manuscript could be identified in the lack of a thorough discussion of the complexity of phasor-FLIM data interpretation in terms of metabolic shifts. Then, more in detail, we agree that the framework of NADPH data interpretation based on the assumption that bound/free NAD(P)H is linearly related to the oxidative phosphorylation/glycolysis ratio - although consolidated in the literature - might not satisfactorily take into account the high complexity

of intracellular biochemistry and the differences among cell types. Thus, we constructively addressed the Reviewer's request for a more careful and critical discussion and interpretation of the current data.

To start, we thoroughly revised the 'abstract', 'introduction' and 'results' sections to remove any premature link to a direct relationship between bound/free NADPH and oxphos/glycolysis. In the revised sections we now refer to collected data for what they strictly represent: a measurement of the bound/free balance of NADPH in the cell (extrapolated from the position of FLIM data along the "metabolic segment" in the phasor plot). Then, we added an entirely new "Discussion" section of the manuscript which addresses the complexity of potential interpretations of the observed shifts in the NADPH bound/free balance in alpha and beta cells. For the sake of clearness, we summarize the key points of this section here:

- 1) In any given cell, beta or alpha, there's a specific biochemical asset made of an array of enzymes, whose identity and level of expression do not significantly change during each of our measurements (that last a few tens of seconds). As a direct consequence, in any given FLIM measurement, we probe the NADPH bound/free ratio maintained by the specific biochemical asset of the cell and the "stationary state" of the corresponding metabolic reactions.
- 2) Based on what said at point 1, at 2.2 mM glucose-concentration we are probing (averaged over the few tens of seconds of the measurement) the specific NADPH bound/free ratio maintained by cellular biochemistry at that specific concentration of the initial substrate; then, upon shifting to 16.7 mM glucose, the NADPH bound/free ratio rearranges itself to a new value, more unbalanced towards the bound NADPH form with respect to the starting condition. The cell biochemistry, in terms of enzyme identity and level of expression, is the same in the two conditions tested; what is changing, as revealed by the new NADPH bound/free ratio, is the balance or stationary state to which the relevant reactions (those involving NADPH) settle down upon increasing the substrate concentration. This in turn reflects, presumably, the activation or increased involvement of specific enzymes. In beta cells, for instance, increased Complex I activity may be responsible of the increased NADPH bound/free ratio observed upon glucose stimulation.
- 3) Based on what is said at points 1 and 2, in presence of a different array of enzymes (i.e. a different biochemical asset of the cell) we may expect a different NADPH bound/free ratio and a different behavior, for instance, in response to shifts in substrate concentration. In this regard, we recently demonstrated that non-secretory cells show indeed a different response to glucose stimulation as compared to secretory beta cells (Ferri et al. FASEB 2020, now cited in the revised text). Based on these considerations, we do agree with the Reviewer that the peculiarities of beta cells (for instance, the cited high levels of glycerol 3-phosphate dehydrogenase expression) make these cells distinct from alpha cells and from any other type of cell. As a consequence, we do also agree that interpreting current data on NADPH bound/free shifts under the same framework (e.g. linearity with oxphos/glycolysis) for both beta and alpha cells might lead to inaccurate conclusions. Thus, in the revised manuscript we now discuss, as suggested by the Reviewer, that a similar increase in the NADPH bound/free ratio may reflect different biochemical activities in beta and alpha cells, that in turn reflect their differences in the expression of key metabolic enzymes. For instance, alpha cells are expected to use much more lactate dehydrogenase as compared to beta cells, and this pathway may contribute to generating bound NADPH in alpha cells upon glucose stimulation.
- 4) Concerning more specifically Reviewer's point 2, we agree with the reviewer that an overall increase in the metabolic fluxes in beta cells as compared to alpha is not entirely unexpected, given the high commitment of beta cells to produce ATP and the differences in key enzymes, as suggested. Such commitment prompts us to speculate that the interpretation of beta cells shifting towards an increase in oxidative phosphorylation upon glucose stimulation may be

acceptable as well as in line with current literature (Wang et al Comm Biol 2021). Conversely, as explained above, the differences of alpha cells in terms of glucose-induced increments in ATP and expression of key metabolic enzymes determine

3. Fig. 5. I do wonder whether the number of subjects here (4) is really enough to draw a robust conclusion about the relationship between glycolysis/ox phos and insulin secretion? Confounding factors (age, sex, BMI, ethnicity etc) of the donors may well play a role? A much larger data set would really be needed for one to have confidence in the conclusion drawn.

We thank the reviewer for this interesting comment. We will respond to this question in separate points.

I do wonder whether the number of subjects here (4) is really enough to draw a robust conclusion about the relationship between glycolysis/ox phos and insulin secretion? A much larger data set would really be needed for one to have confidence in the conclusion drawn.

We agree with the reviewer: more subjects would be needed for robust conclusions at the population level, thus results in Figure 5 (now Figure 4) should only be seen as considerations able to pave the way to further investigations, and we warn the readers in the revised text. As now better specified in the revised text, however, the main focus of the paper is at the single cell level and not at the population level. Concerning this latter, in fact, due to restricted availability of human samples, it would be impractical to obtain a sufficient number of samples in a suitable amount of time such that they are representative of an entire population. Overall, the employed sampling technique (i.e. the so-called *convenience sampling*), together with the restricted number of samples, may lead to biased conclusions. On the other hand, we believe it is very unlikely that four data points (which have been obtained by averaging a huge amount of single-cell responses) lie on a straight line with $R^2=0.98$ just by chance. In addition, the collected four points quite satisfactorily span the distribution curve of Secretary Indexes, which come from separate studies on a heterogeneous population.

Confounding factors (age, sex, BMI, ethnicity etc) of the donors may well play a role?

Of course, it is likely that such factors may play a role, but regarding the presented data we have two considerations. First, the recruited donors do not display high variability (they are not young, have similar BMI, belong to the same ethnicity and have the same sex), thus our conclusions may be true only in the investigated range of factors. The second consideration regards the number of features collected and the proposed model: since the proposed model has only one variable that adequately explains the data variability alone, we would infer that the other factors may play a secondary role, at least in the range of features considered here.

As an exemplary case, you can find in the following the results about the investigation of BMI correlation with both Insulin Secretary Index and overall metabolic response.

Minor

1. Figure 1 and text at the end of the introduction. This would be more logically located in the Results section.

Thanks for this suggestion. The figure was relocated in the Results section as suggested and the text revised accordingly.

2. Fig 2. This is rather similar to Fig 1 from Ref 28. Perhaps relocate to a supplementary section?

Fig. 2 serves as calibration of the measurement using a simplified biological system, INS-1E cells, and actually recapitulates (we agree with the Reviewer) what was shown by Wang et al in a previous publication. We welcomed the suggestion of moving it to supplementary section

Why does the title refer to “alpha/beta” cell lines. INS1E cells are generally considered a beta cell line (vastly more insulin than glucagon expression)

Sorry, our mistake. It has been corrected in the revised version. The new title is: “Metabolic imaging of INS-1E cells”

3. Why was 2.2 mM glucose chosen as the basal concentration? Seems low!

Sorry for the confusion on this point. Islets (and cells) were transiently moved from a “maintenance” glucose concentration of 5,5 mM to 2,2 mM (and here measured) just before glucose stimulation. These details are now specified in the revised text.

4. The Abstract makes a valiant attempt to describe a complex data set but is ultimately confusing. I am not sure that statements like “discordant for high insulin secretion power” will be understood by most readers. Please try to simplify and make more direct.

We appreciate that the Reviewer recognized the effort, especially in the abstract, to describe a complex matter and make it understandable even by non specialized readers. However, we do understand the exhortation to be more explicit and direct. The revised abstract (and text in general) goes in this direction.

5. Line 114 – preliminarLY?

Done, thank you

6. Decimal points should be “.” not “,” throughout

Done, thank you

REVIEWERS' COMMENTS:

Reviewer #1 (Remarks to the Author):

The authors answered my comments and concerns. At this point the paper can be accepted for publication.

Reviewer #2 (Remarks to the Author):

The authors provide a thorough response to my critiques - thank you.

Just a couple of minor points to correct:

1. p11, line 265: "By" the same reasoning (rather than "along"?)
2. p12, line 281,282: Mammalian LDH doesn't use "NADP as a cofactor". It uses NADH, and this as a cosubstrate (with pyruvate).